# Democratizing Reasoning Ability:
# Tailored Learning from Large Language Model

**Zhaoyang Wang**[† 1]  **Shaohan Huang**[2]  **Yuxuan Liu**[† 3]  **Jiahai Wang**[* 1]  **Minghui Song**[2]
**Zihan Zhang**[2]  **Haizhen Huang**[2]  **Furu Wei**[2]  **Weiwei Deng**[2]  **Feng Sun**[2]  **Qi Zhang**[2]

School of Computer Science and Engineering, Sun Yat-sen University, Guangzhou, China[1]
Microsoft[2]    Peking University[3]
wangzhaoy22@mail2.sysu.edu.cn   yx.liu@stu.pku.edu.cn
{shaohanh,zihzha,fuwei,hhuang,zhang.qi}@microsoft.com
wangjiah@mail.sysu.edu.cn

## Abstract

Large language models (LLMs) exhibit impressive emergent abilities in natural language processing, but their democratization is hindered due to huge computation requirements and closed-source nature. Recent research on advancing open-source smaller LMs by distilling knowledge from black-box LLMs has obtained promising results in the instruction-following ability. However, the reasoning ability which is more challenging to foster, is relatively rarely explored. In this paper, we propose a tailored learning approach to distill such reasoning ability to smaller LMs to facilitate the democratization of the exclusive reasoning ability. In contrast to merely employing LLM as a data annotator, we exploit the potential of LLM as a reasoning teacher by building an interactive multi-round learning paradigm. This paradigm enables the student to expose its deficiencies to the black-box teacher who then can provide customized training data in return. Further, to exploit the reasoning potential of the smaller LM, we propose self-reflection learning to motivate the student to learn from self-made mistakes. The learning from self-reflection and LLM are all tailored to the student's learning status, thanks to the seamless integration with the multi-round learning paradigm. Comprehensive experiments and analysis on mathematical and commonsense reasoning tasks demonstrate the effectiveness of our method. The code will be available at https://github.com/Raibows/Learn-to-Reason.

## 1 Introduction

Large language models (LLMs) with emergent abilities have achieved remarkable success across a wide range of tasks, deeply changed the landscape of both research and applications in natural language processing (Brown et al., 2020; Chen et al.,

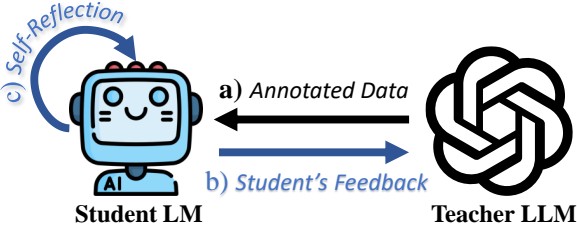

Figure 1: Tailored learning from LLM. In contrast to previous works merely adopt a), we propose b) and c) to further improve the reasoning distillation.

2021; Chowdhery et al., 2022; OpenAI, 2023). And Wei et al. (2022a,b) argue that emergent abilities particularly in reasoning only exist in LLMs whose parameters are commonly larger than 100B. Nevertheless, a line of research (Touvron et al., 2023a,b; Taori et al., 2023; Zeng et al., 2023) has indicated that smaller LMs with about 7B parameters after supervised fine-tuning such as Vicuna (Chiang et al., 2023) can be comparable to LLMs in following human instructions, while still falling short of reasoning. In this paper, we aim to harness the untapped reasoning potential of smaller LMs to democratize this important emergent ability.

Chain-of-Thought (CoT) prompts LMs to generate intermediate reasoning steps (i.e., rationale) to reach the final answer, significantly improving the complex reasoning ability (Wei et al., 2022b; Kojima et al., 2022a; Chung et al., 2022; Wang et al., 2023a). However, it is challenging to prompt smaller LMs to generate reasoning steps, since such ability appears to be exclusive to LLMs (Wei et al., 2022a,b; Chowdhery et al., 2022), which indicates the necessity of utilizing data annotated with rationales to cultivate smaller LMs' reasoning ability. Unfortunately, most existing reasoning datasets lack high-quality rationale annotations, and manual labeling them can be costly. Inspired by the success of collecting instruction data from LLMs (e.g., ChatGPT) for instruction tuning

---

† Work done during internship at Microsoft.
* Corresponding author.

smaller LMs (Wang et al., 2023b; Taori et al., 2023; Touvron et al., 2023a,b), we propose to leverage the rationales generated by LLMs to train smaller LMs to learn to use CoT towards reasoning.

Recently, teaching smaller LMs towards reasoning with the help of LLMs has gained increasing attention. Most of these works (Ho et al., 2023; Magister et al., 2023; Fu et al., 2023b; Shridhar et al., 2023) can be summarized in two main steps: (1) employing LLMs to generate rationales for annotating the training data. (2) Fine-tuning smaller LMs on these data to enable reasoning with CoT. This approach can be viewed as a distant variant of black-box knowledge distillation (Jianping et al., 2021). However, these methods only employ LLMs to annotate the data for training smaller LMs, without leveraging the smaller LMs to assist LLMs in return. As a consequence, the LLMs are not aware of the weaknesses of the smaller LMs, thereby hindering their powerful ability to analyze and provide targeted feedback, which undermines the effectiveness of the reasoning distillation.

To this end, we propose a multi-round interactive learning paradigm to exploit the potential of black-box LLM as a reasoning teacher. In each round of learning, the student (i.e., smaller LM) first provides its learning status to the teacher LLM who then can provide customized rationales as the feedback to the student. The data annotated with these rationales serves as our customized training data. Such a paradigm is natural as it is in inline with how we human beings learn from teachers.

Beyond learning from the teacher, another crucial paradigm for human learning lies in self-reflection on self-made mistakes. In parallel, recent studies (Huang et al., 2022; Shinn et al., 2023; Madaan et al., 2023; Pan et al., 2023) have also shown that LLMs can self-improve by reflecting on their own mistakes. Therefore, we exploit the reasoning potential of smaller LM by eliciting it to take self-reflection on the mistakes. These mistakes can complement correct rationales collected from the teacher LLM to teach the student LM to distinguish bad and good reasoning steps, thereby enhancing its reasoning ability.

Putting them together, as briefly presented in Fig. 1, we propose a tailored multi-round learning paradigm based on the student's learning status and deficiencies, including learning from LLM's customized training data and self-reflection. In summary, our contributions are three-fold:

1) A multi-round learning paradigm is introduced to enable the student LM to provide feedback to the teacher LLM who then can offer customized training data in response, building the interaction between smaller LM and black-box LLM.

2) We propose self-reflection learning that motivates the student to learn from mistakes. Together with learning from customized training data, it can be seamlessly integrated into the multi-round learning paradigm.

3) Experiments and analysis on mathematical and commonsense reasoning tasks demonstrate the effectiveness of our method in distilling the reasoning ability from LLMs to smaller LMs.

## 2 Related Work

**Emergence in LLM** LLMs show emergent abilities in a wide range of NLP tasks (Brown et al., 2020; Chowdhery et al., 2022; Wei et al., 2022a,b; OpenAI, 2023), among which the reasoning ability is the most noteworthy as it requires the model to perform multi-hop reasoning like human beings. Smaller LMs ($< 100B$) are often considered to be falling significantly short in reasoning, highlighting the superiority of LLMs in this aspect (Wei et al., 2022a). In this paper, we aim to democratize such emergent reasoning ability to smaller LMs.

**CoT Prompting** CoT prompts LMs to solve reasoning tasks by generating intermediate rationales to reach the answer, which has greatly improved the reasoning performance (Wei et al., 2022b; Kojima et al., 2022b; Wang et al., 2023a). However, according to the reasoning performance curve (Wei et al., 2022a), the CoT reasoning performance of smaller LMs is far from satisfactory, since the generation of rationales is challenging for them. Chung et al. (2022) reveal that smaller LMs can partially master the CoT skill by training on data with rationales. We show that the CoT performance of smaller LMs can be further improved via tailored learning from LLM's customized training data and self-reflection.

**Distilling Knowledge from LLM** Fine-tuning smaller LMs to follow instructions with high-quality data collected from LLMs shows the feasibility of distilling knowledge from LLMs (Taori et al., 2023; Chiang et al., 2023; Xu et al., 2023). This procedure can also be viewed as a distant variant of black-box distillation (Hinton et al., 2015; Jianping et al., 2021). However, these works aim to improve the instruction-following ability of smaller

LMs, while the reasoning ability that we focus on is often overlooked. Some recent studies (Ho et al., 2023; Fu et al., 2023b; Shridhar et al., 2023) propose to employ LLMs to annotate rationales for training smaller student LMs towards reasoning, not considering the student's feedback to the teacher. In contrast, we exploit the potential of the black-box LLM as the teacher instead of the data annotator by proposing a multi-round learning paradigm. This paradigm enables the mutual feedback between the LLM and smaller LM, thus can make the teacher LLM offer training data tailored for the student LM's learning status. Besides, we propose self-reflection learning to motivate the student LM to learn from mistakes.

## 3 Method

As shown in Fig. 2, we propose a multi-round learning paradigm that motivates the student LM and the teacher LLM to learn feedback from each other in an interactive manner. Specifically, each round of learning consists of three key steps: (1) The student LM undergoes an "exam" on the training set for collecting mistakes which are the wrong generated rationales. Existing works (Fu et al., 2023b; Ho et al., 2023; Shridhar et al., 2023; Magister et al., 2023) merely provide the sample question for the LLM to collect annotated rationales, neglecting the importance of the student's feedback. However, the student's feedback is crucial in knowledge distillation (Fu et al., 2021; Pham et al., 2021; Ren et al., 2023). (2) Therefore, we propose to curate a prompt integrated with the student's wrong rationale to ask the teacher LLM to generate customized feedback for the student. (3) In the last step, the student learns to reason via training on the tailored training data collected from the LLM, and self-reflection on its self-made mistakes. These steps are iterated to improve the reasoning ability of the student LM until convergence.

### 3.1 Undertaking an Exam

Given a dataset $D_{\text{train}} = \{(x, y)\}$, where $x$ is the question and $y$ is the answer, the correct rationale $r$ is often not provided. During inference of CoT, the input is the question $x$, and the student LM's generated output $f(x) = [\hat{r}, \hat{y}]$ is the concatenation of the generated rationale $\hat{r}$ and answer $\hat{y}$. The answer is often at the end of the output.

The student LM undertakes an "exam" on the training set $D_{\text{train}}$ for evaluating the learning sta-

tus, and collecting the mistakes $D_{\text{neg}}$ which are the samples with wrong rationales and answers[1]:

$$D_{\text{neg}} = \{(x, \hat{r}, \hat{y}) \mid \hat{y} \neq y, (x, y) \in D_{\text{train}}\}, \quad (1)$$

for each question, we collect up to 4 wrong rationales through the decoding with sampling strategy. The collected mistake set $D_{\text{neg}}$ reflecting the student's learning status and weakness are used for the following two purposes:

(1) As the feedback for the teacher LLM to generate rationales tailored for the student.

(2) As the negative contrastive samples for the student to learn from self-reflection.

### 3.2 Student's Feedback to LLM

We expect the black-box LLM to be a reasoning teacher instead of a data annotator. Thus, we propose to provide the student's feedback to help the teacher LLM generate customized training data to effectively target the student's weakness. In detail, we devise a prompt template $T$ shown in Fig. 3, which integrates both the question $x$ and the student's feedback (i.e., the wrong rationale $\hat{r}$). The student's feedback can not only (1) assist teacher in identifying deficiencies in student's reasoning, but also (2) as the wrong demonstration example to help LLM increase the chance of generating correct rationales. Besides, to improve the LLM's accuracy and reduce the costs of calling APIs, we follow Zelikman et al. (2022) by adding a hint to explicitly tell LLM the golden answer of the question.

For each sample $(x, \hat{r}, \hat{y}) \in D_{\text{neg}}$, we request the LLM with $T(x, \hat{r}, \hat{y})$ to generate 4 rationales, and only those containing correct answers are retained, since training with diverse reasoning paths can boost the reasoning performance of smaller LMs (Ho et al., 2023; Fu et al., 2023b). The collected rationale together with its question and answer is denoted as $(x, r, y)$, which extends the original data to the customized training data $D_{\text{train}}$.

### 3.3 Tailored Learning

The reasoning ability of student LM $f$ can be improved via tailored learning from both self-reflection and teacher's customized training data.

**Learning from Self-Reflection**  We propose to learn from the mistakes $D_{\text{neg}}$ to simulate the self-reflection process of humans, which can help the

---

[1]Following most existing works, we simply judge the quality of the generated rationale by the correctness of its answer.

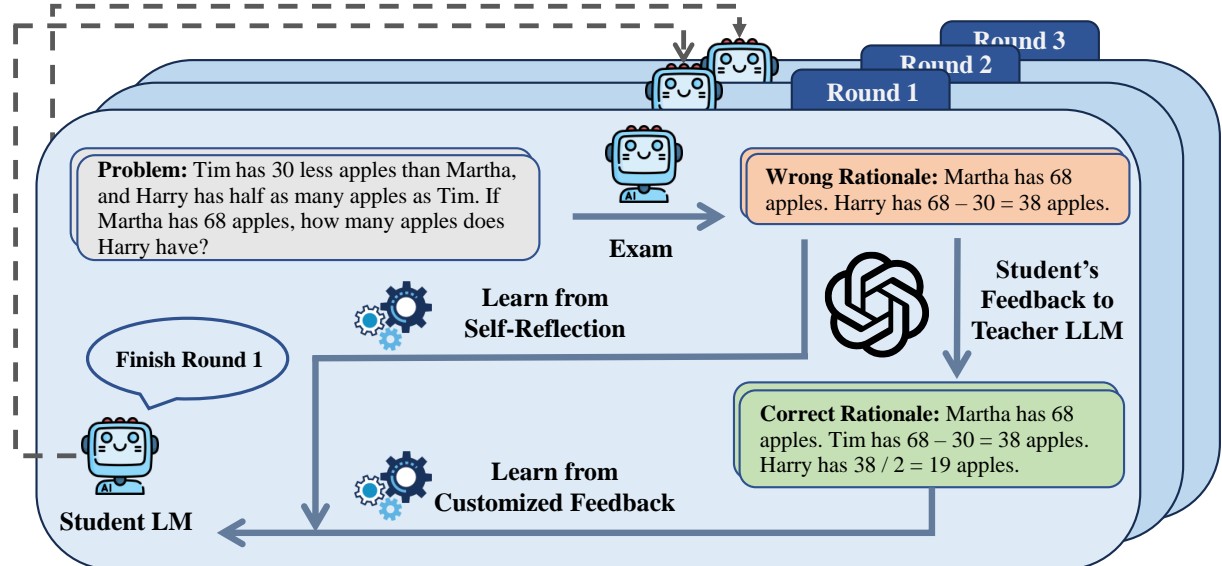

Figure 2: Overview of the proposed multi-round learning paradigm. (1) The student LM first undertakes an "exam" to gather mistakes (i.e., wrong rationales) made by itself. (2) These mistakes are subsequently utilized as the student's feedback to the teacher LLM, which in turn can generate training data (i.e., correct rationales) as the teacher's customized feedback to the student. (3) Finally, the student learns to improve reasoning via self-reflection on self-made mistakes, and assimilation of the customized training data from the teacher LLM. The trained student LM will initiate the next round of learning by repeating the three steps until the performance plateau is reached.

**Question:** ... How man apples does Harry have?
**Wrong Solution:** Bob got 9 oranges…
**Please correct the wrong solution by using better reasoning steps.**
**Hint:** The final answer should be 19.
**Better Reasoning:**

Figure 3: The prompt template $T$ for asking the teacher LLM to generate customized rationales. The part colored in golden is the integrated student feedback.

student LM to identify the quality of different rationales. The utilization can be defined in multiple forms (e.g., likelihood ranking), here we adopt a simple triplet-loss to encourage the model to learn different representations for good and bad rationales. Specifically, the wrong reasoning path $[x, \hat{r}, \hat{y}] \in D_{\text{neg}}$, and the correct reasoning path $[x, r', y] \in D_{\text{train}}$ are utilized as the negative and positive contrastive samples, respectively. The hidden state of the last token is used as the representation of the whole reasoning path, which is denoted as $h_x^{(r,y)}$. Finally, the form of self-reflection learning is defined as follows:

$$\mathcal{L}_{\text{cl}} = \mathbb{E}_{D_{\text{train}}} \max \left\{ 0, \rho - \cos(h_x^{(r,y)}, h_x^{(r',y)}) + \cos(h_x^{(r,y)}, h_x^{(\hat{r},\hat{y})}) \right\}, \quad (2)$$

where $\cos$ denotes the cosine similarity function, and $\rho$ set to 1.0 is the margin. $(x, r, y) \in D_{\text{train}}$ is the anchor sample whose positive and negative samples are randomly sampled from $D_{\text{train}}$ and $D_{\text{neg}}$ with the same question $x$, respectively[2].

**Learning from Customized Feedback** LLM's generated rationales are tailored to the student's weakness, thanks to the previous student's feedback. These collected rationales merged into the training set $D_{\text{train}}$ as the customized feedback for the student, which is used to fine-tune the student LM $f$. In addition, we add several fixed demonstrations "demo" listed in Table 15 to the prefix of each input sample, since recent research (Min et al., 2022; Zelikman et al., 2022; Fu et al., 2023b) have shown that training with demonstration examples can improve the in-context learning ability of LMs. The training objective is as follows:

$$\mathcal{L}_{\text{lm}} = \mathbb{E}_{D_{\text{train}}} \log P_f \left( [\text{demo}, x, r, y] \right), \quad (3)$$

where the square brackets represent the string concatenation. This process can directly help the student LM learn to generate intermediate reasoning steps and master the CoT skill.

[2]Recall that we collect up to 4 unique correct and wrong rationales for each question in $D_{\text{train}}$ and $D_{\text{neg}}$, respectively.

**Algorithm 1** Multi-round learning paradigm.

---

**Require:** the student LM $f$, the teacher LLM, the training data $D_{\text{train}}$, the template $T$ in Fig. 3
 1: Initialize $f^0$ with pre-trained weights and set the learning round count $r \leftarrow 0$
 2: **repeat**
 3:     $r \leftarrow r + 1; f^r \leftarrow f^{r-1}$
 4:     Infer on $D_{\text{train}}$ with $f$ and collects the mistakes $(x, \hat{r}, \hat{y}) \sim D_{\text{neg}}$ by Eq. (1)
 5:     **if** $r \leq 1$ **then**
 6:         Collect the rationale $r$ for each sample of $D_{\text{train}}$ from teacher LLM with $T(x, \text{null}, y)$
 7:     **else**
 8:         Collect the rationale $r$ for each sample of $D_{\text{neg}}$ from teacher LLM with $T(x, \hat{r}, y)$
 9:     **end if**
10:     Optimize weights of $f^r$ using Eq. (4)
11: **until** Converges

---

**Joint Learning** The final optimization incorporates the learning from both self-reflection and LLM's customized feedback. The contrastive learning loss in Eq. (2) and the language modeling loss in Eq. (3) are combined as follows:

$$\mathcal{L} = \mathcal{L}_{\text{lm}} + \lambda \mathcal{L}_{\text{cl}}, \tag{4}$$

where $\lambda$ controls the impacts of self-reflection learning, balancing the two learning objectives.

### 3.4 Multi-round Learning

As depicted in Fig. 2, we adopt a multi-round learning paradigm to iteratively cultivate the reasoning ability of the student LM. Multiple rounds of learning can assist the teacher LLM in staying updated on the student's learning status, and thus offer more customized training data. Based on the student's learning status, the customized training data and self-made mistakes are adjusted in each round and tailored to the student's specific deficiencies.

The untrained student LM nearly has no reasoning ability, resulting in the noisy generations which are unhelpful as the feedback to the teacher LLM. Consequently, to prepare the data required by the initial round, we directly request the teacher LLM to generate rationales for the entire training set excluding the noisy feedback from the student. In the subsequent rounds, we adhere to the procedures outlined in Sections 3.1 to 3.3: (1) the student LM takes an "exam" to reveal self deficiencies and collect mistakes. (2) The teacher LLM is requested to generate customized training data based on the student's feedback. (3) The student is trained via learning both from self-reflection and teacher's customized feedback. These steps are repeated until the student's performance reaches a plateau. The whole paradigm is summarized in Algorithm 1.

## 4 Experiments

### 4.1 Tasks & Datasets

**Mathematical Task** We adopt three math word problem datasets to evaluate the mathematical reasoning ability. GSM8k is a primary school level mathematical dataset (Cobbe et al., 2021). MultiArith is a multi-step arithmetic reasoning dataset (Roy and Roth, 2015). SVAMP is created by applying chosen variations over examples sampled from existing datasets (Patel et al., 2021).

**Commonsense Task** We use two closed-ended question answering datasets to evaluate the commonsense reasoning ability. CSQA (Talmor et al., 2019) is a multi-choice commonsense question answering dataset. StrategyQA dataset (Geva et al., 2021) which implicitly requires reasoning steps and strategies to answer the yes-no questions.

### 4.2 Models & Baselines

**Models** Following previous works (Ho et al., 2023; Zelikman et al., 2022; Hu et al., 2023), we mainly utilize a publicly available LM GPT-J (Wang and Komatsuzaki, 2021) as our student LM which has about 6B parameters. Considering the pricing and availability, we select ChatGPT[3], a popular black-box 175B LLM provided by OpenAI, as our teacher LLM.

**Baselines** To demonstrate the effectiveness of our method, we compare with the following baselines: (1) the teacher LLM and student LM (w/o fine-tuning), for showing the effectiveness of distilling reasoning ability from the LLM. (2) Methods without the help of LLMs, including the student fine-tuned to directly generate answers without rationales, and STaR (Zelikman et al., 2022) which self-iteratively trains the LM to generate rationales and answers with very few annotated data. They are compared to highlight the importance of high-quality rationales in teaching smaller LMs. (3) Three concurrent works which all use LLMs to help train smaller LMs to reason, including LM fine-tuned on CoT data (Magister et al., 2023), Specializing smaller LMs for mathematical reasoning (Fu et al., 2023b), and the LLM-adapter (Hu et al., 2023) which utilizes adapters for efficiently

---

[3]https://chat.openai.com/chat. Most experiments are conducted between February and April of 2023.

| Method | Distillation | CoT | # Params | Mathematical Reasoning | | | Commonsense Reasoning | |
|---|---|---|---|---|---|---|---|---|
| | | | | GSM8K | MultiArith | SVAMP | CSQA | StrategyQA |
| Teacher LLM | No | Yes | 175B | 62.2 | 95.5 | 78.0 | 76.0 | 68.6 |
| Student (w/o Fine-tuning) | No | No | 6B | 2.7 | 9.0 | 20.7 | 34.5 | 47.2 |
| Student (w/ Fine-tuning) | No | No | 6B | 7.2 | 18.0 | 32.3 | 66.7 | 63.9 |
| STaR (Zelikman et al., 2022) | No | Yes | 6B | $10.7^*$ | 53.9 | 26.7 | $\textbf{72.5}^*$ | 60.0 |
| LLM-Adapter (Hu et al., 2023) | Yes | Yes | 6B | $10.6^*$ | $79.2^*$ | $45.0^*$ | - | - |
| Specializing (Fu et al., 2023b) | Yes | Yes | 11B | $27.1^*$ | $63.0^*$ | $35.6^*$ | - | - |
| CoT Fine-tuned (Magister et al., 2023) | Yes | Yes | 11B | $18.4^*$ | - | - | - | $63.8^*$ |
| One-Round Distillation | Yes | Yes | 6B | 15.6 | 81.5 | 47.7 | 68.1 | 63.8 |
| + Multi-round | Yes | Yes | 6B | $32.0_{+16.4}$ | $83.1_{+1.6}$ | $51.3_{+3.6}$ | $70.2_{+2.1}$ | $65.5_{+1.7}$ |
| + Self-Reflection | Yes | Yes | 6B | $\textbf{33.1}_{+1.1}$ | $\textbf{85.4}_{+2.3}$ | $\textbf{55.0}_{+3.7}$ | $71.3_{+1.1}$ | $\textbf{65.9}_{+0.4}$ |

Table 1: Accuracy (%) on various reasoning tasks with different methods. "LLM-Adapter" refers to results of GPT-J using LoRA adapter (Hu et al., 2022). "Specializing" refers to results of FlanT5-XXL (Chung et al., 2022) which has about 11B parameters. "CoT Fine-tuned" refers to results of T5-11B (Raffel et al., 2020) fine-tuned on CoT data from GPT-3 175B (Brown et al., 2020). $^*$ denotes the results are from the original paper. Indentation means the modifications are based on the up-level indentation. The best performance among small LMs are marked in **bold**.

tuning on CoT data. (4) Our one-round distillation method, for demonstrating the superiority of the proposed multi-round learning paradigm.

## 4.3 Experimental Setup

The student is fine-tuned with a learning rate of $1e-6$ in 10 epochs using AdamW (Loshchilov and Hutter, 2019) in default. Without any heavy tuning, $\lambda$ in Eq. (4) is set to $0.5$ to control the impact of self-reflection. The CoT prompt accompanied by a fixed 3-shot demonstration is used for most datasets to balance the efficiency and performance. Some prompts are referred to previous research (Zelikman et al., 2022). And we use greedy decoding to generate the rationale and answer for evaluation. More implementation details are in Appendix A.

## 4.4 Main Results

The evaluation results are presented in Table 1.

**Effect of Distillation** From the results of smaller LM with or without distillation, it is evident that the reasoning performance of smaller LM can be significantly improved by distilling the reasoning ability from LLM. Although the student LM falls short in mathematical reasoning, it can achieve comparable performance in commonsense reasoning with the teacher LLM while being 20x smaller in size.

**Importance of Rationales** CoT can significantly improve reasoning performance which shows the necessity of high-quality rationales in teaching smaller LMs. Though STaR performs well in CSQA which often only involves single-step reasoning, the self-generated rationales encounter dif-

ficulties when applied to other multi-step reasoning tasks. Conversely, nearly all distillation methods can beat STaR in mathematical reasoning, which indicates that LLM's generated rationales can often better guide the smaller LM to reason.

**Comparison with Concurrent Works** Compared to concurrent distillation works (Hu et al., 2023; Fu et al., 2023b; Magister et al., 2023), our method consistently achieves better performance across all datasets, which demonstrates the success of customized feedback from the black-box LLM. For GSM8K, in contrast to training an 11B model with 130k rationales used by Specializing, our method can yield better performance with a 6B model and only 54k rationales, significantly reducing the cost of model training and data collection.

**Effect of Multi-round & Self-reflection** Compared with our one-round distillation approach, multi-round learning leads to an average improvement of $5.1$ in accuracy, indicating the success of building the interaction between teacher and student. Besides, the self-reflection learning can further exploit the reasoning potential of the student LM. Another advantage is that the self-reflection can be seamlessly integrated into multi-round learning, pushing the boundaries of reasoning.

## 5 Analysis

### 5.1 Contribution of Student's Feedback

To validate the contribution of student's feedback to LLM, an ablation study is conducted by removing this feedback of the requesting prompt template

| GSM8K Example#1 | Ground Truth: 196 |
|---|---|
| Question | Olaf has 150 cars at first. Grandpa gave Olaf twice as many toy cars as the uncle. Dad gave Olaf 10 toy cars, 5 less than Mum. Auntie gave Olaf 6 toy cars, 1 more than the uncle. How many toy cars does Olaf have in total, finally? |
| Student's Feedback | Olaf has 150 toy cars. His family gives him twice as many toy cars as the uncle. The uncle gives Olaf 10 toy cars. Dad gives Olaf 10 toy cars. Auntie gives Olaf 6 toy cars. So, Olaf has 150 + 10 + 10 + 6 = 236 toy cars. |
| Teacher's Response | Dad: 10 cars, 5 less than Mum; Mum: 10 + 5 = 15 cars; Aunt: 6 cars, 1 more than uncle; Uncle: 6 - 1 = 5 cars; Grandpa: twice as many toy cars as the uncle, so 5 * 2 = 10 cars; Total: 150 + 15 + 10 + 6 + 5 + 10 = 196 cars. |
| Teacher's Response w/o Feedback | Dad: 10 cars; Mum: 10 + 5 = 15 cars; Uncle: 6 - 1 = 5 cars; Grandpa: 2 * 5 = 10 cars; Total: 150 cars + 10 cars + 15 cars + 5 cars + 10 cars = 190 cars. |

| StrategyQA Example#1 | Ground Truth: Yes |
|---|---|
| Question | Can you write a whole Haiku in a single tweet? |
| Student's Feedback | Haiku is a form of poetry that has three lines and a 5-7-5 syllable count. A tweet is 140 characters long. So, the answer is No. |
| Teacher's Response | A haiku can be written in a single tweet if it follows the 5-7-5 syllable pattern within 140 characters. For example, "Autumn moonlight. A worm digs silently into the chestnut. Dewdrops on a leaf." The poem is considered a Haiku even if it is shorted, so it's possible to fit a whole Haiku in a single tweet. So, the answer is Yes. |
| Teacher's Response w/o Feedback | A Haiku is a traditional form of poetry that is comprised of three lines. The first line is five syllables, the second line is seven syllables and the last line is five syllables. So, one cannot write a whole Haiku in a single tweet as it exceeds the character limit. So, the answer is No. |

Table 2: Case study of the impact of student's feedback on the teacher LLM's response. The incorrect reasoning steps and answers are colored in red. More cases are listed in Appendix C.

| Dataset | # Request | # Success | Accuracy |
|---|---|---|---|
| GSM8K | 5701 | 5250 | 28.2 |
| w/o Feedback | 5701 | 4641 | 26.5 $_{-1.7}$ |
| SVAMP | 168 | 166 | 51.3 |
| w/o Feedback | 168 | 140 | 48.3 $_{-3.0}$ |
| StrategyQA | 328 | 317 | 65.5 |
| w/o Feedback | 328 | 134 | 63.9 $_{-1.6}$ |

Table 3: The effect of student's feedback to the teacher LLM for the 2nd round learning, based on the same 1st round. "w/o Feedback" indicates removing student's feedback in the prompt template shown in Fig. 3. # Request and Success are the number of requests to LLM and response with correct rationales, respectively.

| Dataset | Method | Distance | Preference |
|---|---|---|---|
| GSM8K | Student | 51.00 | 73.63 |
| | + Self-Reflection | 65.08 | 79.11 |
| SQA | Student | 5.03 | 96.54 |
| | + Self-Reflection | 24.78 | 98.91 |

Table 4: Comparison of the student LM with and without self-reflection learning on GSM8K and SQA datasets. "Distance" measures the Euclidean distance between correct and wrong reasoning paths in latent space. "Preference" is the likelihood ratio of correct reasoning paths to wrong ones. Both are higher is better.

(Fig. 3). Results in Table 3 show that student feedback to LLM can first help the teacher LLM to generate more accurate and tailored rationales (larger # Success), which is then beneficial to the student's learning (higher Accuracy). Note that cooperating with our multi-round learning paradigm, the cumulative gains of student's feedback can be substantial. Further, we take a case study of the teacher LLM's generated rationales in Table 2 which shows that the LLM can often response improved rationales when the student's feedback is taken into account. For StrategyQA, the teacher LLM even gives a counterexample to the student's wrong answer, indicating the LLM can provide customized training data based on the student's feedback.

## 5.2 Effect of Self-Reflection

First, to intuitive understand the effect of self-reflection learning, Fig. 4 visualizes the latent space representations of generated rationales. It shows that the self-reflection could effectively cluster correct rationales and wrong ones respectively, helping the model to distinguish each other. Moreover, we compare the distance and preference differences in Table 4 which indicates that the self-reflection contributes to aligning the preference of the student

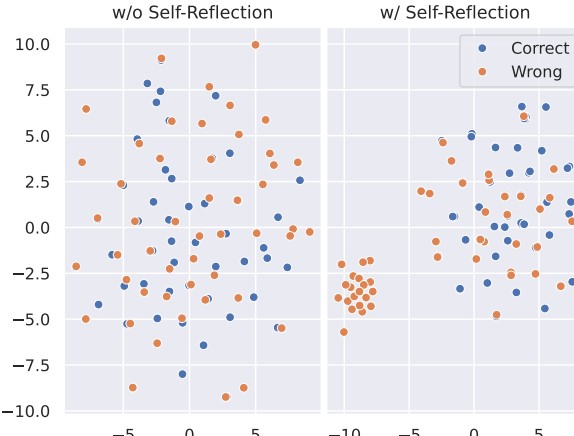

Figure 4: The t-SNE visualization (van der Maaten and Hinton, 2008) of latent space representations of rationales generated on the GSM8K dataset.

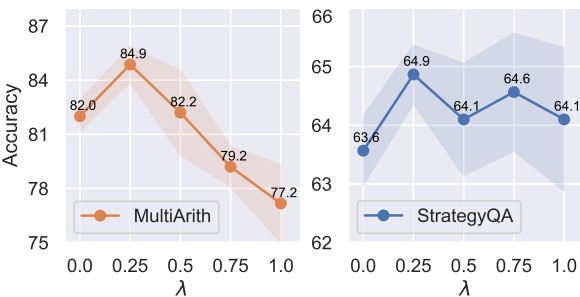

Figure 5: The effect of $\lambda$ in Eq. (4) on the initial round performance of the student LM. $\lambda = 0.0$ indicates the absence of self-reflection learning.

LM with correct reasoning paths, while away from self-made wrong ones.

Fig. 5 illustrates the effect of the self-reflection learning on the reasoning performance. The observation is consistent with findings in Table 1 that self-reflection learning can help improve the reasoning ability when $\lambda < 0.5$. However, excessive emphasis on self-reflection learning (i.e., a larger value of $\lambda$) typically leads to poorer performance and instability, especially for the MultiArith dataset. We conjecture that it has a negative impact on the learning of teacher's training data.

To verify the above hypothesis, we plot the loss curve in Fig. 6. It shows that the excessive emphasis on self-reflection learning (higher $\lambda$) can result in underfitting of the these training data within a limited number of training steps. Consequently, the reasoning performance of the student LM could be significantly decreased due to not fully converged. In general, a small value of $\lambda$ is preferred to achieve a balanced learning approach that incorporates both the teacher's rationales and self-made mistakes.

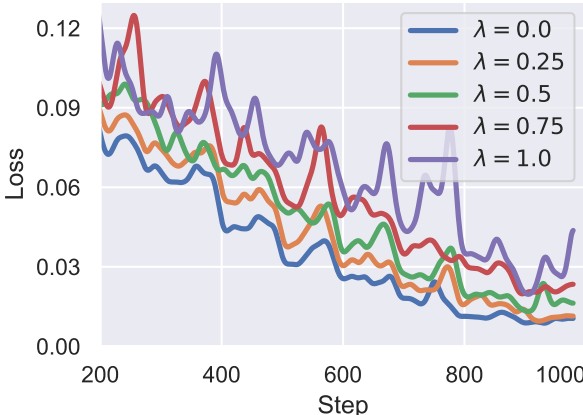

Figure 6: The training loss of Eq. (3) in the initial round of the student LM with different weight $\lambda$ on the Multi-Arith dataset. We also observe that the loss of Eq. (2) with different $\lambda$ can all converge.

## 5.3 Analysis of Multi-round Learning

We examine each learning round of the student LM, as detailed in Table 5. The error rate and accuracy are typically gradually decreased and increased with the learning rounds, respectively. This is because of each round of learning aims to enhance the student LM in solving the questions that were not learned well in previous round. Additionally, inspired by recent research on employing the LLM as the evaluator (Chiang and Lee, 2023; Fu et al., 2023a; Liu et al., 2023), we instruct GPT-4 (OpenAI, 2023) to automatically evaluate the quality of generated rationales. From the results in Table 6, we find that there is an enhancement in the quality of both generated correct rationales and wrong ones as the learning rounds progress. However, the gains in reasoning performance reach a plateau after several rounds of training. This can be attributed as follows: (1) For GSM8K, the most challenging task, the student is reaching its capacity after 3 rounds of learning, still not performing well (49.2 ER). (2) For SVAMP and CSQA, relatively easy tasks, the student achieves a good performance on the training set after the 2nd round, leading to a small ER. Consequently, the prepared data for the next round will be relatively scarce, which is unlikely to further help improve the student.

We conduct the 4th round learning on GSM8K for justifying the above analysis, where the ER remains unsatisfactory (51.8 ER) despite a marginal improvement ($+1.4\,\Delta$) in accuracy. Besides, the results of the 3rd round on SVAMP and CSQA datasets show that there are no more gains after the 2nd round. Thus, we suggest to take early stopping

| Dataset | | Initial | 1st | 2nd | 3rd |
|---|---|---|---|---|---|
| GSM8K | # Data | - | 15k | 16k | 13k |
| | ER | 98.3 | 76.3 | 66.2 | 49.2 |
| | Acc/$\Delta$ | 2.7 | +12.9 | +12.6 | +2.4 |
| SVAMP | # Data | - | 2k | 0.6k | 0.3k |
| | ER | 76.0 | 24.0 | 16.7 | 17.6 |
| | Acc./$\Delta$ | 20.7 | +27.0 | +3.6 | +1.0 |
| CSQA | # Data | - | 26k | 7k | 3k |
| | ER | 67.8 | 18.9 | 7.6 | 9.2 |
| | Acc./$\Delta$ | 34.5 | +31.8 | +3.9 | -0.6 |

Table 5: Observation of the student LM in each round of learning. "Initial" refers to model w/o distillation. "#Data" represents the size of training samples. "ER" refers to the error rate on train set. "Acc" denotes the initial accuracy of the student LM, and "$\Delta$" indicates its performance change after each round.

| Dataset | Round | Correct | Wrong |
|---|---|---|---|
| GSM8K | Initial | $2.59_{\pm0.27}$ | $1.02_{\pm0.07}$ |
| | 1st | $4.50_{\pm0.18}$ | $1.15_{\pm0.20}$ |
| | 2nd | $4.88_{\pm0.14}$ | $1.26_{\pm0.23}$ |
| SVAMP | Initial | $4.53_{\pm0.20}$ | $1.07_{\pm0.18}$ |
| | 1st | $4.86_{\pm0.16}$ | $1.09_{\pm0.21}$ |
| | 2nd | $4.90_{\pm0.24}$ | $1.11_{\pm0.20}$ |
| CSQA | Initial | $4.44_{\pm0.22}$ | $1.24_{\pm0.28}$ |
| | 1st | $4.84_{\pm0.27}$ | $1.41_{\pm0.28}$ |
| | 2nd | $4.96_{\pm0.12}$ | $1.55_{\pm0.33}$ |

Table 6: Results of GPT-4 score for student LM's generated rationales in each round of learning. The score is given based on accuracy and quality of the reasoning path. "Correct" and "Wrong" stand for the rationales with correct answers and wrong answers, respectively.

in the multi-round learning if the student can nearly reach its plateau. By prior estimation of the task difficulty and observing performance gains in each round, we can avoid excessive parameter tuning on the number of learning rounds and balance the reasoning performance and training costs.

### 5.4 Feasibility Study

To further benefit the community concerning about individual affordable computation resources, we conduct a feasibility study by using different LMs spanning from 760M to 2.7B parameters. The tested models include two common LM architectures, i.e., encoder-decoder and decoder-only. The results shown in Table 7 first suggest that the reasoning abilities of these small LMs can all be en-

| | Method | 760M | 770M | 1.3B | 2.7B |
|---|---|---|---|---|---|
| SVAMP | Student | 0.0 | 2.7 | 5.3 | 3.7 |
| | + Distillation | 11.0 | 13.3 | 31.7 | 34.3 |
| | + Self-Reflection | $14.7_{+3.7}$ | $15.3_{+2.0}$ | $32.0_{+0.3}$ | $36.3_{+2.0}$ |
| | + Multi-round | $15.3_{+0.6}$ | $17.0_{+1.6}$ | $35.0_{+3.0}$ | $36.0_{-0.3}$ |
| SQA | Student | 0.0 | 39.6 | 51.2 | 38.9 |
| | + Distillation | 62.0 | 62.2 | 62.0 | 62.2 |
| | + Self-Reflection | $64.0_{+2.0}$ | $64.2_{+2.0}$ | $64.8_{+2.8}$ | $65.2_{+3.0}$ |
| | + Multi-round | $64.8_{+0.8}$ | $62.4_{-1.8}$ | $65.8_{+1.0}$ | $63.8_{-1.4}$ |

Table 7: Results of our method with various LM sizes. "760M", "770M", "1.3B" and "2.7B" refer to T5-Large (Raffel et al., 2020), GPT-2 Large (Radford et al., 2019), OPT-IML (Iyer et al., 2023) and GPT-Neo (Gao et al., 2020; Black et al., 2021), respectively. The indentation means the modifications are based on the up-level indentation.

hanced with the proposed self-reflection learning. With self-reflection, student LMs often achieve satisfying performance with just one round of learning for commonsense tasks. Moreover, we find that our multi-round learning can generally further improve the performance in mathematical reasoning. However, there are no more gains for StrategyQA, as it heavily relies on the memorization of commonsense knowledge mostly acquired from the pre-training stage, rather than on complex reasoning. Another evidence is that increasing the model size seems not to have contribution to the performance on this dataset. Besides, the relatively limited capacity of these smaller LMs may also restrict the gains from additional rounds of learning.

## 6 Conclusion

In this paper, we propose a tailored learning approach to cultivate the reasoning ability of the smaller LM, aiming to democratize the emergent reasoning ability of the LLM. First, we propose a multi-round interactive learning paradigm that enables the teacher LLM to provide customized training data according to the student's feedback. Next, we propose the self-reflection learning to motivate the student to distinguish correct rationales from wrong ones. Further, the integration of learning from LLM's customized feedback and self-reflection can complement each other within the proposed multi-round learning paradigm. The empirical results from mathematical and commonsense reasoning tasks demonstrate the success of unleashing the reasoning potential of smaller LMs. We believe that these findings can benefit the open-source and NLP communities in the era of LLM.

## Limitations

In this section, we discuss the limitations of our method with integrity while offering potentially useful advice for future research.

1) Our experiments primarily utilize ChatGPT and GPT-J (Wang and Komatsuzaki, 2021) as the teacher LLM and student LM, respectively, due to the considerations of availability and costs. Although fine-tuning GPT-J on the outputs of ChatGPT boosts their reasoning performance, a substantial gap still remains between them. It is valuable to validate our findings using more powerful LMs (e.g., LLaMA (Touvron et al., 2023a,b)). And training better foundation LMs should be the primary task for the open-source community, since imitating proprietary LLMs may be a false promise (Gudibande et al., 2023).

2) We have demonstrated the importance of student's feedback in distilling the knowledge from the black-box LLM, but without extensive engineering the feedback prompt templates (e.g., explicitly instructing the LLM to act as a teacher). And the interactions (e.g., use reinforcement learning to connect LLM and smaller LM) can be explored in the future.

3) Our self-reflection learning currently is defined in a straightforward triplet-loss form. However, the core of self-reflection is learning from mistakes. Thus, the training objectives or forms can be defined in various ways, such as ranking loss or verbal critic are expected to further help the smaller LMs to reflect and learn from mistakes.

4) Evaluating the correctness of generated rationale is mainly based on the final answer. Though most existing works (Zelikman et al., 2022; Ho et al., 2023; Fu et al., 2023b; Shridhar et al., 2023) in this field adopt this simple criterion, we call attention to develop more trustworthy criteria to evaluate the quality of rationales. Potential methods can be using GPT-4 (OpenAI, 2023) or a process reward model (Lightman et al., 2023) for automatic evaluation.

## Ethics Statement

**Risk in using closed-source LLMs** Though the datasets used for evaluation is publicly available, the annotated rationales in this paper are collected from close-source ChatGPT provided by OpenAI.

Open-source LLMs (e.g., LLaMA) have boomed in recent months, it is noteworthy that many of them use the outputs from closed-source LLMs (e.g., Alpaca and Vicuna are trained on ChatGPT's outputs) for further improvements. According to the Sec. 2 "Usage Requirements", within OpenAI's terms of use[4], there exists a prohibition against "use output from the Services to develop models that compete with OpenAI". However, beyond its terms of use, the crucial matter lies in determining "ownership of the copyright pertaining to the outputs of generative AI". As of today, there remains an ambiguity regarding the copyright status of generative AI outputs, both in scholarly circles and legal contexts. Compelling evidence indicates that these closed-source LLMs undergo training using numerous copyrighted materials, such as books, academic publishings, etc. Thus, we think at least the authors of the training data that directly supports LLM's outputs hold the copyright, as opposed to the LLM service provider. The prompt creators may also hold the copyright if their prompts substantially influence LLM's outputs. For open-source and research communities, we call for a responsible discussion about data collection.

**Social Impact** This paper explores how to utilize the LLM as a teacher to enhance the reasoning performance of smaller LMs, which can help democratize these emergent abilities for the benefit of broader communities (e.g., math education). Furthermore, we firmly believe that the utilization of LLMs can be a significant area of interest in natural language processing applications and research.

## Acknowledgements

We thank the anonymous reviewers for their insightful and valuable comments. This work is supported by the National Natural Science Foundation of China (62072483), and the Guangdong Basic and Applied Basic Research Foundation (2022A1515011690, 2021A1515012298).

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

## A   Implementation Details

The codes will be made publicly available after anonymous reviewing period.

### A.1   Data Preparation

The dataset statistics are shown in Table 8. Following Ho et al. (2023), the data of SVAMP (Patel et al., 2021), MultiArith (Roy and Roth, 2015) and StrategyQA (Geva et al., 2021) is split with a ratio of $70 : 30$ for the training and evaluation, while GSM8K (Cobbe et al., 2021) and CSQA (Talmor et al., 2019) datasets follow the original split. In mistakes collection, we use sampling decoding to prompt student LM to generate 4 rationales for each sample, and only the wrong ones are collected. In rationales collection, the teacher LLM is requested to generate 4 diverse rationales for each question, and only the correct ones are collected. An example of Fig. 3 for using student's feedback to request the LLM is shown in Table 12. The decoding generation configs are listed in Table 9.

### A.2   Training & Evaluation

**Hyperparameter**   Experiments are performed with the help of Transformers[5] (Wolf et al., 2020) and Deepspeed[6] (Rajbhandari et al., 2020) libraries. We use 8 Tesla V100 GPUs with FP16 for training and evaluation. The adopted training hyperparameter settings across all datasets are shown in Table 10. The student LM is trained with a $1e{-}6$ learning rate for the initial round learning, and $7e{-}7$ for the following rounds, to make the training more stable. And we set a random seed 42 for all experiments to ensure reproducibility.

**Demonstration**   Following Min et al. (2022); Zelikman et al. (2022); Fu et al. (2023b), we use several fixed demonstrations selected from the training set as the prefix of each sample to improve the in-context learning performance. Considering the memory consumption and efficiency, we use 3-shot demonstrations for GSM8K, MultiArith, and SVAMP datasets. For CSQA and StrategyQA, we respectively use 5-shot and 4-shot demonstrations to reduce the label bias (Zhao et al., 2021) since they are essentially 5 ("a, b, c, d, e") and 2 ("yes, no") labels classification tasks. These demonstrations are listed in Table 15.

---

[5] https://github.com/huggingface/transformers
[6] https://github.com/microsoft/DeepSpeed

---

| Dataset | Type | # Train | # Test | Split |
|---|---|---|---|---|
| GSM8K | Mathmatical | 7473 | 1319 | Original |
| MultiArith | Mathmatical | 420 | 180 | 70:30 |
| SVAMP | Mathmatical | 700 | 300 | 70:30 |
| CSQA | Commonsense | 9741 | 1221 | Original |
| StrategyQA | Commonsense | 1603 | 687 | 70:30 |

Table 8: Dataset statistics.

| Arguments | Mistakes | LLM |
|---|---|---|
| Temperature | 1.0 | 1.0 |
| Top-p | - | 0.9 |
| Top-k | 50 | - |
| Max Generation Len. | 128 | 128 |
| # Return Sequences | 4 | 4 |

Table 9: Generation configs for collecting student's self-made mistakes and rationales from teacher LLM.

| Hyperparameter | Value |
|---|---|
| Epoch | 10 |
| Batch Size | 16 |
| Learning Rate | $\{1e{-}6, 7e{-}7\}$ |
| $\beta$ of AdamW | $(0.9, 0.999)$ |
| $\epsilon$ of AdamW | $1e{-}8$ |
| Weight Decay | 0.01 |
| Warmup Steps | 100 |

Table 10: Training hyperparameter settings.

In addition, from pilot experiments, we empirically find that assigning less weights (0.1) to the fixed demonstration examples than the input sample helps the model focus on the input sample and yield better performance, which can be investigated in the future.

**Evaluation**   We use a simple-yet-effective CoT prompt template as follows:

$$\text{Question: } x \text{ \textbackslash n Reasoning: } r \text{ \textbackslash n Answer: } y \quad (5)$$

where \n is the line break symbol, $x$ is the question, $r$ and $y$ are expected reasoning steps and answer, respectively. The greedy decoding is adopted for the generation of the student LM though beam search may further improve the performance. The answer extraction of evaluation is simply using the first valid token after the "Answer:", which can avoid complex post-processing.

## B  Generalization Results

Generalization experiments are conducted to evaluate the generalization of the student LM, as shown in Table 11. The results reveal the following insights: (1) the in-domain generalization performance is enhanced after the reasoning distillation, while the out-of-domain (OOD) performance is usually slightly decreased. This finding is consistent with Fu et al. (2023b) although our method is better than theirs in terms of OOD performance. (2) The in-domain performance can be further improved by employing our multi-round learning paradigm. And we surprisingly find that, for some cases, the OOD performance can also be improved via multi-round learning. This can be attributed to that the customized training data of the following rounds may assists the model in generalizing its reasoning abilities to other domains. (3) The student LM trained on the GSM8K dataset exhibits the most significant improvements in in-domain reasoning performance. Note that the GSM8K dataset is the most challenging one among these mathematical datasets. Consequently, it is reasonable to expect gains on the other datasets if the student can already tackle the difficult problems.

## C  Case Study

**Contribution of Student's Feedback**  Additional examples of the LLM's generated rationales are presented in Table 13. We observe that the teacher LLM, ChatGPT, is capable of generating more detailed and precise reasoning steps when provided with student's feedback (i.e., wrong solution). These detailed reasoning steps can help address the student's deficiencies and thereby improve the reasoning performance in the subsequent round of learning. Although both rationales, with and without feedback, are correct, their quality can vary. More precise and customized rationales can help the student better understand its own mistakes, especially coupled with our self-reflection learning, which is beneficial for student's reasoning learning.

**Multi-round Learning**  To better understand the impact of each learning round, we conduct a case study in Table 14. First, it is clear that the student LM initialized with pre-trained weights (i.e., the $0^{th}$ round) is powerless to generate meaningful answers for the mathematical reasoning task, which may confuse the teacher LLM. Thus, we tend not to utilize these noisy feedback for preparing the train-

ing data of the initial round. Second, the LLM's generated response is often tailored to student's current deficiencies, thus effectively improving student's reasoning performance in the next round of learning. Third, a single round of distillation may not enable the student to solve challenging questions. However, with the help of our multi-round learning paradigm, the student can have the opportunity to tackle such challenging questions.

| Train on | | Evaluation on | | | | |
|---|---|---|---|---|---|---|
| | | GSM8K | MultiArith | SVAMP | CSQA | StrategyQA |
| None | | 2.7 | 9.0 | 20.7 | 34.5 | 47.2 |
| GSM8K | 1st | 15.6 | 46.6 | 25.3 | 28.4 | 38.3 |
| | Last | 32.0 | 80.3 | 42.3 | 30.0 | 38.3 |
| MultiArith | 1st | 4.7 | 81.5 | 14.7 | 32.3 | 52.5 |
| | Last | 5.0 | 83.1 | 19.3 | 31.4 | 52.1 |
| SVAMP | 1st | 4.0 | 12.4 | 47.7 | 29.6 | 45.4 |
| | Last | 5.4 | 14.6 | 51.3 | 34.0 | 44.7 |
| CSQA | 1st | 2.6 | 5.1 | 12.3 | 68.1 | 48.0 |
| | Last | 2.3 | 5.1 | 14.3 | 70.2 | 51.1 |
| StrategyQA | 1st | 3.8 | 9.0 | 19.0 | 33.3 | 63.8 |
| | Last | 9.6 | 9.6 | 17.3 | 33.5 | 65.5 |

Table 11: Generalization results of our multi-round learning paradigm on in-domain and out-of-domain reasoning tasks. "None" refers to the student model initialized with pre-trained weights, without any fine-tuning. "1st" refers to the student LM taken the first round of learning. Similarly, "Last" refers to the last round of learning. Results colored in green indicate the in-domain reasoning, while blue indicate the out-of-domain reasoning.

---

Question: Is Europa linked to Viennese waltzes?
Wrong Solution: Europa is a moon of Jupiter, and it is named after a Phoenician queen of Crete. Viennese waltzes are a type of dance that originated in Austria. So, the answer is No.
Please correct the wrong solution by using better reasoning steps.
Hint: The final answer should be yes.
Better Reasoning: While Europa is a moon of Jupiter and named after a Phoenician queen of Crete, it is also the title of a famous waltz composed by Austrian composer Johann Strauss II in 1889. The waltz is named after the mythical princess Europa from Greek mythology, who was abducted by Zeus and taken to Crete. So, the answer is Yes.

Question: Would a honey badger fit inside an oven?
Wrong Solution: Honey badgers are large animals that can grow up to be about the size of a small dog. An oven is a small, enclosed space. So, the answer is No.
Please correct the wrong solution by using better reasoning steps.
Hint: The final answer should be yes.
Better Reasoning: The answer is actually Yes, a honey badger could fit inside an oven, but the size of a honey badger can vary and an oven can also vary in size. So, the answer is yes.

Question: Would a pacifist be opposed to hunting?
Wrong Solution: Pacifists believe in non-violence and do not support the use of violence. Hunting is a form of violence. So, the answer is Yes.
Please correct the wrong solution by using better reasoning steps.
Hint: The final answer should be no.
Better Reasoning: While pacifists do believe in non-violence, their opposition is generally focused on human-on-human violence. Some pacifists may be opposed to hunting due to their belief in non-violence towards animals, but it is not necessarily a universal position among pacifists. So, the answer is No.

Question: { question $x$ }
Wrong Solution: { mistake $\hat{r}$ }
Please correct the wrong solution by using better reasoning steps.
Hint: The final answer should be { answer $y$ }.
Better Reasoning: { Expected LLM's customized feedback }

Table 12: An example of the prompt template introduced in Fig. 3 for asking the LLM to generate customized rationales, for a given question $x$, student's mistake $\hat{r}$, and the ground truth answer $y$ from the StrategyQA dataset.

| | GSM8K Example#2    Ground Truth: 12 |
|---|---|
| Question | Janet bought some muffins at the bakery. Each muffin is 75 cents. Janet paid 20 and got 11 in change back. How many muffins did Janet buy? |
| Student's Feedback | Janet bought some muffins at the bakery. Each muffin is 75 cents. Janet paid 20 and got 11 in change back. We know that:1) Janet bought some muffins at the bakery.2) Each muffin is 75 cents.3) Janet paid 20 and got 11 in change back.From these three pieces of information, we can infer that: Janet bought 9 muffins. |
| Teacher's Response | We know that Janet paid 20 and got 11 in change back. To find how much money Janet paid to buy muffins, we can subtract 11 from 20 which results in 9. If 1 muffin costs 75 cents and Janet paid 9 then she must have bought 12 muffins ( 9 divided by 75 cents per muffin = 12 muffins). |
| Teacher's Response w/o Feedback | The change Janet got back is 11. This means Janet spent 20 - 11 = 9 on muffins. If each muffin costs 75 cents, then Janet bought 9 / 0.75 = 12 muffins. |

| | SVAMP Example#1    Ground Truth: 5 |
|---|---|
| Question | A waiter had 11 customers. After some left he still had 3 customers. How many more customers left than those that stayed behind? |
| Student's Feedback | To find the number of customers who left, we need to subtract the number of customers who stayed behind from the number of customers who left. The number of customers who left is 11 - 3 = 8. |
| Teacher's Response | Let X be the number of customers who left the restaurant. We know that 11 - X = 3, as 3 customers stayed behind. We can solve for X by adding X to both sides of the equation, resulting in X = 11 - 3 = 8. Therefore, the number of customers who left is 8 and the number who stayed is 3. 8 - 3 = 5 is the number of customers who left than those that stayed behind. |
| Teacher's Response w/o Feedback | To find the number of customers who left, we must subtract the number of customers who stayed from the total number of customers the waiter had: 11 - 3 = 8. Therefore, 8 more customers left than those that stayed behind. |

| | SVAMP Example#2    Ground Truth: 9 |
|---|---|
| Question | 5 children were riding on the bus. At the bus stop 63 children got off the bus while some more got on the bus. Then there were 14 children altogether on the bus. How many more children got on the bus than those that got off? |
| Student's Feedback | To find the number of children that got on the bus, we need to subtract the number of children that got off the bus from the total number of children on the bus. Therefore, the number of children that got on the bus is 14 - 63 = -49. |
| Teacher's Response | Let X be the number of children who got on the bus. Initially, there were 5 children on the bus and 63 children got off. Therefore, after getting off the bus, the number of children on the bus is 5 - 63 = -58. Adding the children that got on the bus, we get -58 + X = 14. Therefore, X = 72. And we can calculate the difference between children that got on the bus and children that got off. 72 - 63 = 9. Therefore, 9 more children got on the bus than those that got off. |
| Teacher's Response w/o Feedback | First, we know that 5 children were riding on the bus before the bus stop, then 63 children got off the bus. So, there were 5 - 63 = -58 children on the bus. Then, 14 children were on the bus, so we know that 14 - (-58) = 72 children got on the bus. To find the number of children who got on the bus than those who got off, we find the difference between the number of children who got on and the number who got off which is 72 - 63 = 9 children. |

| | StrategyQA Example#2    Ground Truth: Yes |
|---|---|
| Question | Do mail carriers need multiple uniforms? |
| Student's Feedback | Mail carriers work for the United States Postal Service (USPS) and they have one uniform. So, the answer is No. |
| Teacher's Response | Mail carriers typically have multiple uniforms, with some designed for indoor use, some for outdoor use, and others for different weather conditions. Some mail carriers may also have different clothing items for specialized tasks such as carrying mail bags. The answer is Yes. |
| Teacher's Response w/o Feedback | Mail carriers work outdoors in various weather conditions. They need to wear different clothes to be protected from different weather conditions. So, the answer is Yes. |

Table 13: Case study of the impact of student's feedback on the teacher LLM's response. The incorrect reasoning steps and answers are colored in red. With the student's feedback, the teacher can often generate more detailed and precise reasoning steps.

| GSM8K Example#1 |
| --- |

**Question:** The number of short students in a class is 2/5 of the total number of students. If there are 90 tall students, and the class has 400 students, calculate the total number of students with average height. **Ground Truth: 150**

| Round | Student's Generation    Correctness: ✗ |
| --- | --- |
| 0th | The number of short students is 2/5 of the total number of students.The number of tall students is 2/5 of the total number of students. The number of students with average height is 2/5 of the total number of students. |
| | Teacher's Feedback |
| 1st | The number of short students in the class is 2/5 of the total number of students, which means there are (2/5) * 400 = 160 short students. If there are 90 tall students, then the number of students with average height can be calculated by subtracting the number of short and tall students from the total number of students. The total number of students with average height is 400 - 160 - 90 = 150 students. |
| | Student's Generation    Correctness: ✗ |
| 1st | There are 90 tall students in the class. There are 2/5 of 400 students, which is 2/5 * 90 = 36 short students. So the total number of students in the class is 90 + 36 = 126. |
| | Teacher's Feedback |
| 2nd | There are 90 tall students in the class. Short students are 2/5 of the total number of students. The total number of students in the class is 400. Short students: 2/5 * 400 = 160. We know the total number of students in the class is 400, so 90 + 160 = 250 is the number of tall and short students. Therefore, the total number of students with average height is 400 - 250 = 150. |
| | Student's Generation    Correctness: ✗ |
| 2nd | The number of short students in a class is 2/5 of the total number of students. So, 2/5 * 400 = 160. The number of tall students in a class is 90. So, the total number of students with average height is 400 - 160 - 90 = 170. |
| | Teacher's Feedback |
| 3rd | The total number of students in the class is 400. The number of short students is 2/5 of the total number of students, 2/5 * 400 = 160. The number of tall students is 90. The total number of short and tall students is 160 + 90 = 250. So the total number of students with average height is 400 - 250 = 150. |
| | Student's Generation    Correctness: ✔ |
| 4th | The number of short students is 2/5 of the total number of students, which is 2/5 * 400 = 160. The total number of students with short height is 160 + 90 = 250. The total number of students with average height is 400 - 250 = 150. |

| CSQA Example#1 |
| --- |

**Question:** Where do students keep rubber bands?
(a) trees (b) drug store (c) trash (d) beside table (e) classroom **Ground Truth: (e)**

| Round | Student's Generation    Correctness: ✗ |
| --- | --- |
| 0th | The answer should be a place where students would keep rubber bands. Bedside tables are often where students keep their rubber bands. Therefore, the answer is bedside table (d). |
| | Teacher's Feedback |
| 1st | The answer should be a place where students might keep rubber bands. Students might keep rubber bands in their desks in the classroom. Therefore, the answer is classroom (e). |
| | Student's Generation    Correctness: ✔ |
| 1st | The answer should be a place where students would keep rubber bands. Rubber bands are often used for school projects, so students would keep them in their classroom. Therefore, the answer is classroom (e). |

Table 14: Case study of the student LM's generation and teacher LLM's feedback by multi-round learning. The teacher can generate more tailored training data thanks to being aware of the student's deficiencies. And the student can gradually improve its reasoning ability with the help of these customized feedback.

| 3-shot demonstrations of GSM8K dataset |
| --- |

**Question:** Albert is wondering how much pizza he can eat in one day. He buys 2 large pizzas and 2 small pizzas. A large pizza has 16 slices and a small pizza has 8 slices. If he eats it all, how many pieces does he eat that day? \n **Reasoning:** A large pizza has 16 slices, so 2 large pizzas have 32 slices. A small pizza has 8 slices, so 2 small pizzas have 16 slices. If we add 32 slices and 16 slices, we get 48 slices. Therefore, he will eat 48 slices of pizza in that day. \n **Answer:** 48

**Question:** Mary does her grocery shopping on Saturday. She does her shopping only at a specific store where she is allowed a credit of 100, which must be paid in full before her next shopping trip. That week she spent the full credit limit and paid 15 of it on Tuesday and 23 of it on Thursday. How much credit will Mary need to pay before her next shopping trip? \n **Reasoning:** Mary spends her entire credit limit of 100 on Saturday. On Tuesday, she pays 15 towards her debt. On Thursday, she pays 23 towards her debt. This leaves her with a remaining balance of 100 - 15 - 23, which is equal to 62. \n **Answer:** 62

**Question:** Ralph is going to practice playing tennis with a tennis ball machine that shoots out tennis balls for Ralph to hit. He loads up the machine with 175 tennis balls to start with. Out of the first 100 balls, he manages to hit 2/5 of them. Of the next 75 tennis balls, he manages to hit 1/3 of them. Out of all the tennis balls, how many did Ralph not hit? \n **Reasoning:** Ralph hits 2/5 of the first 100 balls, so he hits 40 balls. Then, Ralph hits 1/3 of the next 75 balls, so he hits 25 more balls. In total, Ralph hits 40 + 25 = 65 balls. Finally, we know that Ralph started with 175 balls, so 175 - 65 = 110 balls not hitted. \n **Answer:** 110

| 3-shot demonstrations of MultiArith dataset |
| --- |

**Question:** There are 64 students trying out for the school's trivia teams. If 36 of them didn't get picked for the team and the rest were put into 4 groups, how many students would be in each group? \n **Reasoning:** The number of students who got picked for the team is 64 - 36 = 28. To find how many students would be in each group, we need to divide the number of students by the number of groups, which is 28 / 4 = 7. \n **Answer:** 7

**Question:** Cody bought 7 boxes of chocolate candy and 3 boxes of caramel candy. If each box has 8 pieces inside it, how much candy did he have total? \n **Reasoning:** First, we need to find the total number of boxes Cody bought, which is 7 + 3 = 10 boxes. Then, we can multiply the number of boxes by the number of pieces of candy in each box to find the total amount of candy. Therefore, Cody had 10 x 8 = 80 pieces of candy in total. \n **Answer:** 80

**Question:** For Halloween Robin scored 23 pieces of candy. She ate 7 pieces the first night and then her sister gave her 21 more pieces. How many pieces of candy does Robin have now? \n **Reasoning:** We need to add the number of pieces of candy she had after the first night to the number of pieces her sister gave her. Therefore, the total number of pieces of candy Robin has now is 23 - 7 + 21 = 37. \n **Answer:** 37

| 3-shot demonstrations of SVAMP dataset |
| --- |

**Question:** Paul had 50 books. After buying some in a garage sale he had 151 left. How many books did he buy? \n **Reasoning:** The number of books Paul bought can be found by subtracting the final number of books from the initial number of books: 151 - 50 = 101. Therefore, Paul bought 101 books in the garage sale. \n **Answer:** 101

**Question:** Luke played a trivia game and scored 154 points. If he gained the 11 points in each round. How many rounds did he play? \n **Reasoning:** We need to divide Luke's total score by the number of points he gained in each round. Therefore, the number of rounds Luke played is 154 / 11 = 14. \n **Answer:** 14

**Question:** Julia played tag with 17 kids on monday, 15 kids on tuesday and 2 kids on wednesday. How many kids did she play with altogether? \n **Reasoning:** To find the total number of kids Julia played with, we need to add the number of kids she played with on each day. Therefore, the total number of kids Julia played with is 17 + 15 + 2 = 34. \n **Answer:** 34

| 5-shot demonstrations of CSQA dataset |
| --- |

**Question:** What do people use to absorb extra ink from a fountain pen? \n **Answer Choices:** \n (a) shirt pocket \n (b) calligrapher's hand \n (c) inkwell \n (d) desk drawer \n (e) blotter \n **Answer:** The answer must be used to absorb extra ink. Blotters are designed to absorb liquids. Therefore, the answer is blotter (e).

**Question:** What home entertainment equipment requires cable? \n **Answer Choices:** \n (a) radio shack \n (b) substation \n (c) television \n (d) cabinet \n (e) desk \n **Answer:** The answer must require cable. Cable is used to provide satellite channels to televisions. Therefore, the answer is television (c).

**Question:** Sammy wanted to go to where the people were. Where might he go? \n **Answer Choices:** \n (a) populated areas \n (b) race track \n (c) desert \n (d) apartment \n (e) roadblock \n **Answer:** The answer must be a place with many people. Populated areas, by definition, have a lot of people. Therefore, the answer is populated areas (a).

**Question:** Where do you put your grapes just before checking out? \n **Answer Choices:** \n (a) mouth \n (b) grocery cart \n (c) super market \n (d) fruit basket \n (e) fruit market \n **Answer:** The answer should be the place where grocery items are placed before checking out. Of the above choices, grocery cart makes the most sense for holding grocery items. Therefore, the answer is grocery cart (b).

**Question:** Google Maps and other highway and street GPS services have replaced what? \n **Answer Choices:** \n (a) united states \n (b) mexico \n (c) countryside \n (d) atlas \n (e) oceans \n **Answer:** The answer must be something that used to do what Google Maps and GPS services do, which is give directions. Atlases were also used to give directions. Therefore, the answer is atlas (d).

| 4-shot demonstrations of StrategyQA dataset |
| --- |

**Question:** Are chinchillas cold-blooded? \n **Reasoning:** Chinchillas are rodents, which are mammals. All mammals are warm-blooded. So, the answer is No. \n **Answer:** No

**Question:** Would Janet Jackson avoid a dish with ham? \n **Reasoning:** Janet Jackson follows an Islamic practice. Islamic culture avoids eating pork. Ham is made from pork. So, the answer is Yes. \n **Answer:** Yes

**Question:** Can a honey bee sting a human more than once? \n **Reasoning:** Human skin is tough, and the bee's stinger gets lodged in the skin. The stinger becomes separated from the bee which dies soon after. So, the answer is No. \n **Answer:** No

**Question:** Is average number of peas in a pod enough commas for a billion? \n **Reasoning:** The average number of peas in a pod is 6 or 7. A billion is a number that has only 3 commas in it. So, the answer is Yes. \n **Answer:** Yes

Table 15: The demonstrations used for each dataset. The "\n" indicates a line break. The **key token** is marked in bold for clear view. The prompt for CSQA is slightly different from others since we adopt the original prompt template of STaR (Zelikman et al., 2022). And we only use 5 out of 7 demonstrations from STaR.