# OpenReview forum: "Democratizing Reasoning Ability: Tailored Learning from Large Language Model"
_EMNLP/2023/Conference — EMNLP 2023 Main_

### Official Review · Reviewer_3M1o · 2023-08-03

**Typos Grammar Style And Presentation Improvements:** 1. Line 377
**Soundness:** 4

**Excitement:**

4: Strong: This paper deepens the understanding of some phenomenon or lowers the barriers to an existing research direction.

**Missing References:**

Missing citation / discussion around a recent paper suggesting that distilling from larger models only helps to imitates the larger model but does not give it capabilities - https://arxiv.org/abs/2305.15717


**Paper Topic And Main Contributions:**

This paper proposes to distill reasoning abilities from (closed) larger language models to (open) smaller models, using a multi-round learning procedure, where the student’s incorrect reasoning chains are used to correct/generate reasoning chains from LLMs iteratively. In addition to the data generated from LLMs, the paper uses a ‘self-reflection’ objective which helps a model to contrastively learn from its own incorrect generations. Experimental results show the efficacy of the proposed approach.


**Questions For The Authors:**

1. What do you do with the generated rationales + answer if it’s not easy to automatically parse and identify the answer? E.g. the answer occurs somewhere in the middle instead of at the end.
2. Line 288 — does this include LM loss over the demo too?
3. Table 1 – does the one-round distillation numbers use self reflection?
4. Table 1 – are the missing numbers (‘-’) because the original papers didn’t evaluate on those datasets? Given that some of the baselines are not that hard to setup/run (e.g. finetuning on CoT), it would be useful to have the missing numbers too. Also using GPT-J 6B uniformly (instead of T5 11B in some rows) across the table would be a better setup.
5. Table 3 – while the reduced gains in accuracy across rounds make intuitive sense as mentioned in the paper, another source of the reduced gains could be that total data is less in later rounds. It might be good to control for this by using the same amount of data (e.g. equal to the round with least data) during finetuning.
6. Line 100 – how does the self-reflection objective proposed in the paper compare to the other works which explore similar ideas related to self-reflection?
7. (thought) Could this method be made more efficient by using a different subset of the training data in each round? (e.g. as observed, right now, it seems to plateau quite quickly)


**Reasons To Accept:**

1. Given how important & widespread distillation from closed-source LLMs to smaller LMs has become in the last few months (e.g. Alpaca, Vicuna etc.), extending it to reasoning and using a multi-round feedback setup makes a lot of sense. It’s interesting to see that using the student’s responses + multi-round feedback is able to give significant improvements (table 1).
2. The evaluation and experiments seem reasonably thorough, covering multiple datasets (across two types of tasks – math reasoning and commonsense reasoning) with clear benefits from using a multi-round learning paradigm. (see ‘questions for authors’ and ‘reasons to reject’ section for a few suggested experiments to add)
3. The paper includes ablations to demonstrate the benefit of using student’s feedback (instead of directly prompting LLM to generate CoT), using a multi-round procedure (vs single round) and the self-reflection objective.
4. The paper is well written and easy to read/understand.


**Reasons To Reject:**

1. (minor) Use of annotated reasoning chains –

a. Potential missing experiment which would be good to add - how does the proposed method compare to using manually labeled reasoning chains? (I understand the proposed approach is much cheaper, but it would be interesting to see how much of the gap is closed). E.g. I think StrategyQA does come with reasoning chains which could be used as an upper bound.

b. Line 271 – how do you get the anchor sample? Is it just a randomly sampled (x,r,y) from the LLM where y is correct? Or do you use an annotated reasoning chain (in which case the above baseline becomes more important)?

2. The paper does not discuss the legality/copyright aspect of using outputs from a closed-source model to finetune an open-source model. While the paper does briefly mention it (line 571) it could be good to add details and/or point readers to resources which discuss this.


**Reproducibility:**

4: Could mostly reproduce the results, but there may be some variation because of sample variance or minor variations in their interpretation of the protocol or method.

**Reviewer Confidence:**

4: Quite sure. I tried to check the important points carefully. It's unlikely, though conceivable, that I missed something that should affect my ratings.

---

> ### Author Rebuttal · Authors · 2023-08-29
>
> Thanks for your thorough reviews and insightful comments. We have carefully considered your advice, and we hope the following point-to-point response could address your concerns.
>
> ### W1.a: use of annotated reasoning chains
>
> Thanks for your advice. We think comparing teacher LLM's annotated data with manually annotated ones would be interesting.
>
> 1. As introduced in L60, one of the motivations for collecting LLM's CoT outputs is the lack of high-quality manually annotated datasets. Among the 5 datasets we adopted, only StrategyQA has high-quality fact statements that can be used to form a reasoning path.
>
> 2. We first use GPT-4 (please refer to W4 of Reviewer BrcX for details) to automatically evaluate the quality of the annotated rationales. Then we use manually annotated data to fine-tune the student LM. We show the results of GPT-4 score and accuracy in the following table.
> 3. From the results, we find that
>    - **The quality and correctness of manually annotated reasoning chains are better than ChatGPT's generated ones.** Nevertheless, we could fill this gap by collecting more reasoning paths for each question (i.e., about 3x training data in total rounds), improving the diversity.
>    - **With the proposed multi-round learning paradigm, the fine-tuned student model could achieve better performance than model using manually annotated rationales**, which indicates the success of our method.  Besides, there is a slight preference in generations of our student model by GPT-4.
>
> | Dataset (annotator) | Score of annotated rationale | Acc  | Score of Student's generated correct | Score of student's generated wrong |
> | ------------------- | ---------------------------- | ---- | ------------------------------------ | ---------------------------------- |
> | StrategyQA (manual) | 4.80±0.30                    | 63.8 | 4.13±0.31                            | 1.11±0.22                          |
> | StrategyQA (ours)   | 4.51±0.40                    | 65.9 | 4.24±0.37                            | 1.14±0.23                          |
>
> ### W1.b: how do you get the anchor sample in L271
>
> **The anchor samples and the positive samples are all from the collected teacher LLM's generated rationales**, since all the 5 datasets except StrategyQA do not have high-quality annotated reasoning paths for training, and manually labeling them can be costly. **We would like to revise our paper to explicitly clarify its reliance on the output from the teacher LLM.**
>
>  ### W2: discuss the legality/copyright aspect of using outputs from closed-source LLM
>
> **We really appreciate your advice on enhancing the discussion about legality/copyright of using outputs from closed-source LLMs. We would like to add the following discussion to Sec. Ethics Statement.**
>
> Although open-source LLMs (e.g., LLaMA) have boomed in recent months, it is noteworthy that many of them use the outputs from closed-source LLMs (e.g., Alpaca and Vicuna are trained on ChatGPT's outputs) for further improvements. According to the Sec. 2 "Usage Requirements", within OpenAI's [terms of use](https://openai.com/policies/terms-of-use), **there exists a prohibition against "use output from the Services to develop models that compete with OpenAI".** This stipulation poses a potential quandary for the open-source community.
>
> However, beyond its terms of use, **the crucial matter lies in determining "ownership of the copyright pertaining to the outputs of generative AI"**. As of today, there remains an ambiguity regarding the copyright status of generative AI outputs, both in scholarly circles and legal contexts.
>
> Compelling evidence indicates that these closed-source LLMs undergo training using numerous copyrighted materials, such as books, academic publishings, etc (though OpenAI recently allowed users to block their web crawlers). The LLM, or any other generative AI, essentially models its training data, selectively retrieving tokens, phrases, or even sentences from its memory to formulate responses to users' prompts.  So, in the current landscape of copyright law, we think **at least the authors of the training data that directly supports LLM's outputs hold the copyright, as opposed to the LLM service provider.** The prompt creators may also hold the copyright if their prompts substantially influence the LLM's outputs.
>
> For open-source and research communities, we should **recognize the terms of use of closed-source LLMs and have a responsible discussion about collecting these closed-source LLM's outputs**. Currently, academic research may not directly face copyright risks or lawsuits. Moreover, we need to make progress in building truly open-source and responsible LLMs to reduce the reliance on closed-source generative AI services to benefit human society.
>
> **Also, there are several related materials[1,2,3,4] available for the readers.**
>
> [1] OpenAI. Terms of use. https://openai.com/policies/terms-of-use
>
> [2] Gil Appel, Juliana Neelbauer, and David A. Schweidel. Generative AI Has an Intellectual Property Problem. https://hbr.org/2023/04/generative-ai-has-an-intellectual-property-problem
>
> [3] Gervais, Daniel J. "AI Derivatives: The Application to the Derivative Work Right to Literary and Artistic Productions of AI Machines." *Seton Hall L. Rev.* 52 (2021): 1111.
>
> [4] The Verge media. The scary truth about AI copyright is nobody knows what will happen next. https://www.theverge.com/23444685/generative-ai-copyright-infringement-legal-fair-use-training-data
>
> ### Q1: parse the generated rationales + answers
>
> Most existing works in this research field often use regex to backwardly search the first valid answer with a specific pattern (e.g., continuous digits for the math reasoning task). Our work is also based on this, but make the following improvements to make it more easily:
>
> 1. **A structured prompt template is adopted**. In most cases, the template is
>
>    ```
>    Question: {question} \n Reasoning {rationale}\n Answer: {final answer}
>    ```
>
>    Then, the final answer is extracted by simply matching these pre-defined structures. Please refer L895 in Appendix A.2 for details.
>
> 2. Thanks to the in-context learning of LLMs, **we use few-shot demonstrations (also in the above structured template) to enforce the models to generate structured outputs**. The details about these demonstrations are in L876.
>
> ### Q2: Line 288 — does this include LM loss over the demo too?
>
> Yes, but we generally find given a lower weight (0.1) to LM loss over the demonstration can yield better performance. The details are in L889 of Appendix A.2.
>
> ### Q3: Table 1 – does the one-round distillation numbers use self reflection?
>
> No. **Only the bottom row in Table 1uses self-reflection learning** (i.e., every round of training uses self-reflection). For single round w/ self-reflection, please refer to Table 5.
>
> ### Q4: missing numbers in Table 1
>
> Yes, the missing numbers are because original papers do not evaluate on these datasets. **Actually, the method of Fine-tuning CoT is similar to our one-round distillation despite the difference in collecting data and training.** While we recognize that adopting GPT-J uniformly would yield a better setup, we intend to revise the results accordingly, given the availability of resources. There are two primary reasons hindering implementations of them on other datasets:
>
> 1. To fairly reproduce other baselines on not evaluated datasets, we have to collect the training data from the **specific closed-source LLMs which may not be accessible (e.g., PaLM-540B) or too expensive (e.g., code-davinci-002 is 10x expensive than Turbo-3.5)**. And it is not easy to fine-tune T5-xxl which has 11B (~2x params than GPT-J) for our limited resources.
> 2. **The original training data of some baseline methods are not available.** Thus, we cannot directly use these data to fine-tune GPT-J and report its results. For baseline using the same student model GPT-J (e.g., STaR), we have replicated it on other datasets. To further alleviate this problem and validate the effectiveness of our method on T5, **we take additional experiments with T5-large-760M. The results are in W3 of Reviewer BrcX.**
>
> ### Q5: using the same amount of data for each round of training
>
> We think control for this is good to show the contributions of tailored training data. **However, Table 4 wants to present and analyze the learning and convergence status for each round of learning, i.e., how to estimate and determine the rounds of learning,  which is also an important practical problem for our method.** Recall that in each round of learning, we only collect the questions that student LM currently cannot answer correctly.  As explained in L479,
>
> - For GSM8K, the training data is still a lot but the student LM has reached its capacity limit.
> - For SVAMP and CSQA, the student LM has overfit the training set. Then, for preparing the next round of training, the needed request data is reduced.
>
> Nevertheless, we show the primary results of using the same amount of training data (disjoint subset of the entire set) for each round.
>
> | Dataset & Metric     | Initial | Round 1 | Round 2 |
> | -------------------- | ------- | ------- | ------- |
> | SVAMP # Data         | -       | 0.3k    | 0.3k    |
> | SVAMP ER             | 76.0    | 54.9    | 55.9    |
> | SVAMP ACC. /$\Delta$ | 20.7    | +15.0   | +4.0    |
> | CSQA # Data          | -       | 1k      | 1k      |
> | CSQA ER              | 67.8    | 47.7    | 43.3    |
> | CSQA ACC. / $\Delta$ | 34.5    | +22.6   | +1.4    |
>
> **We find the performance gains in round 2 are still reduced even though the training data has been controlled.**  And the overview performance on both training set (ER) and test set (ACC) has decreased, which indicates that more rounds of learning are needed to reach a plateau. Please refer to Q7 for further discussion.
>
> ### Q6: L100 differences of self-reflection objectives
>
> As stated in L100, recent research works[1,2] on LLMs (not our smaller LMs) have used "self-reflection" to improve LLM's performance. The differences are:
>
> 1. They instruct LLMs (> 100B) to take self-reflection, **while our method focuses on smaller LMs.**
> 2. They typically provide the mistakes (**similar to our 'student feedback to teacher LLM'**) or existing generations in the context, with specific prompts to instruct the LLMs to take self-reaction and improve their generations at the inference. **In contrast, our self-reflection explicitly trains the model to distinguish the good rationales from self-made mistakes via Eq. 2.** Besides, we think such smaller LMs (about 6B params) are not capable of self-reflection via instructions.
>
> **We also analyze the essential role of our self-reflection learning in W1 of Reviewer MSxv.**
>
> [1] Shinn, Noah, et al. "Reflexion: Language agents with verbal reinforcement learning." arXiv preprint arXiv:2303.11366 (2023).
>
> [2] Madaan, Aman, et al. "Self-refine: Iterative refinement with self-feedback." arXiv preprint arXiv:2303.17651 (2023).
>
> ### Q7:  Is it more efficient by using a different subset of the training data in each round?
>
> We think it will have some conflicts with the motivation of the proposed method. From Algorithm 1, the training data for the $(i+1)$ round is decided by:
>
> ```
> How many mistakes are made by the fine-tuned student model of the previous round (i.e., the $i$ round)
> ```
>
> **If we pre-partition the dataset into several subsets for each round, then we cannot fully utilize these mistakes over the entire training set. The mistakes on the specific training subset can be biased.**  And according to Table 4 and primary results in Q5, we think there are two main reasons for the relatively fast plateau:
>
> 1. For GSM8K, a difficult dataset, the student model still has large gains after 2 rounds of learning. The capacity of the student limits its further improvements.
> 2. For other relatively easy datasets, e.g., CSQA, if we use a disjoint subset for training each round, the gains become less, which may lead to a slow plateau but may not improve further than using the entire training set.
>
> ### Q8: missing citation
>
> We will cite this [paper]( https://arxiv.org/abs/2305.15717) and introduce it in introduction and limitations sections to highlight the potential false promises of fine-tuning small models on LLM's outputs. We also plan to investigate it in future work.
>
> ### Q9: typos
>
> Thanks for your careful reviews. We will fix these typos and ungrammatical sentences.
>
> We will revise L38 to explicitly distinguish between the size of LLMs and smaller LMs.
>
> ```
> And Wei et al. (2022a,b) argue that emergent abilities particularly in reasoning only exist in LLMs whose parameters are commonly larger than 100B.
> ```

---

### Official Review · Reviewer_BrcX · 2023-08-04

**Soundness:** 3

**Excitement:**

3: Ambivalent: It has merits (e.g., it reports state-of-the-art results, the idea is nice), but there are key weaknesses (e.g., it describes incremental work), and it can significantly benefit from another round of revision. However, I won't object to accepting it if my co-reviewers champion it.

**Paper Topic And Main Contributions:**

This paper proposes a framework that utilizes the rationales generated by large language models (LLMs) to improve the chain-of-thought (CoT) reasoning ability of smaller LMs. The main difference between prior works that employ LLMs as data generators is that the feedback from student LM is used to assist the teacher LLM to produce customized training data and the student LM itself to take self-reflection. Besides, multi-round learning enables the interaction between teacher LLM and small LM. The proposed method is compared with multiple baselines, including concurrent works (e.g., LM fine-tuned on CoT data, Specializing smaller LMs), on mathematical and commonsense reasoning tasks across five datasets. Experiments demonstrate the effectiveness of the proposed method in distilling the reasoning ability from LLMs to small LMs.

**Reasons To Accept:**

-	The idea of incorporating feedback from student LM into knowledge distillation, facilitating the interaction between teacher and student models, is novel and interesting.
-	Experiments are sufficient, with several competitive baselines and insightful ablation studies. Experimental results demonstrate the superiority of the proposed method over the baselines.


**Reasons To Reject:**

-	One notable limitation pertains to the simplistic criterion employed for evaluating the quality of the generated rationales, which solely relies on the correctness of the final answer. The underlying assumption, where right rationales are associated with right answers and wrong rationales are with wrong answers, may not be true. For example, the generated rationale may be irrelevant to the final answer, no matter whether it is correct or not. In that case, the collected mistakes and training data might be noisy. Sanity check or evaluations can be conducted to filter out inconsistent rationale-label pairs.
-	Although the teacher LLM generates a correct response with the student LM’s feedback, it does not explain or justify the student’s mistake. Considering the overhead of collecting feedback and customized training data, how about prompting teacher LLM to generate rationales based on gold labels and then training the student LM with the generated rationales in multiple rounds? This would be an interesting baseline to take.
-	Only GPT-based models are tested. It would be better to consider other architectures and even smaller models (e.g., T5).
-	No human evaluation is conducted to verify the enhanced reasoning ability of small LM.


**Reproducibility:**

4: Could mostly reproduce the results, but there may be some variation because of sample variance or minor variations in their interpretation of the protocol or method.

**Reviewer Confidence:**

4: Quite sure. I tried to check the important points carefully. It's unlikely, though conceivable, that I missed something that should affect my ratings.

---

> ### Author Rebuttal · Authors · 2023-08-29
>
> We sincerely appreciate your positive evaluation of our research and the insightful advice you've provided. We hope the following point-to-point response would alleviate your concerns.
>
> ### W1:  solely relies on the correctness of the final answer
>
> 1. We acknowledge that the assumption of "correct final answer -> good rationales" is not flawless. However, **most related works[1,2,3,4]** in fine-tuning smaller language models towards reasoning solely rely on this assumption to collect training data. Besides, **most CoT of LLM research works** report their performance based on this assumption though correct final answers do not guarantee correct rationales.
>
> 2. Quality evaluation of language model’s generated output has often been a great challenge and open research topic. Especially when our target - rationales, are more complicated and have a higher demand for logical correctness. **As far as we know, there are no simple automatic evaluation/filter methods to filter inconsistent rationale-answer pairs.**
>    - Also, as stated in L61, **most of these datasets in this area lack high-quality labeled rationales but only have final answers**. Thus, we cannot adopt traditional methods (e.g., BLEU) to assess its quality.
>    - Beyond data-cleaning by verification, we aim to improve the quality of training rationales at the source. We propose a bi-directional framework to enable the student’s feedback to the teacher LLM. We **focus on how to build the interaction between student LMs and the black-box teacher LLM** to provide a high-quality multi-round learning curriculum and obtain significant empirical gains.
>
> 3. We are happy to discuss and adopt effective sanity check methods to reduce potential noise in our training data for further improvement. But note that **the noise data may have little impact on our method**, since results in Table 4 still show consistent improvements as the learning round progresses. **We already adopted the following steps to alleviate this issue:**
>    - We add the hint (providing the golden answer) to request the teacher LLM to generate rationales as the training data, which is shown to be useful in improving the correctness of rationales[5,6].
>    - We use wrong examples (i.e., student's feedback) to motivate the teacher LLM to generate better rationales, which can be viewed as self-correction of LLMs[7,8] (c.f. Table 2)
>    - The data collection and training are in a multi-round fashion, which could gradually reduce the noise (C.f. W4 in the below response).
> 4. Inspired by recent research[9], **one potentially useful method is to use a process reward model** (which still needs labeled data to train) **or GPT-4** (please refer to W4) to score the intermediate reasoning steps (i.e., rationale) and filter out noisy ones. **We would like to discuss this in Sec. Limitations and leave it in the future work** to further improve our method. **Thanks again for your insightful advice.**
>
> [1] Fu, Yao, et al. "Specializing Smaller Language Models towards Multi-Step Reasoning." International Conference on Machine Learning (ICML 2023)
>
> [2] Hsieh et al. "Distilling Step-by-Step! Outperforming Larger Language Models with Less Training Data and Smaller Model Sizes." In Findings of the Association for Computational Linguistics: ACL 2023
>
> [3] Ho et al. "Large Language Models Are Reasoning Teachers."  In the Association for Computational Linguistics: ACL 2023
>
> [4] Magister et al. "Teaching Small Language Models to Reason."  In the Association for Computational Linguistics: ACL 2023
>
> [5] Zelikman, Eric, et al. "Star: Bootstrapping reasoning with reasoning." Advances in Neural Information Processing Systems 35 (2022): 15476-15488.
>
> [6] Zheng, Chuanyang, et al. "Progressive-hint prompting improves reasoning in large language models." arXiv preprint arXiv:2304.09797 (2023).
>
> [7] Shinn, Noah, et al. "Reflexion: Language agents with verbal reinforcement learning." arXiv preprint arXiv:2303.11366 (2023).
>
> [8] Madaan, Aman, et al. "Self-refine: Iterative refinement with self-feedback." arXiv preprint arXiv:2303.17651 (2023).
>
> [9] Lightman, Hunter, et al. "Let's Verify Step by Step." arXiv preprint arXiv:2305.20050 (2023).
>
> ### W2: use data w/o feedback to train student LMs in multiple-rounds
>
> 1. As we introduced in L221-224 and from the empirical observations, the student feedback is useful in two aspects:
>    - **Help the teacher LLM generate more accurate rationales** (improve the efficiency of collecting training data), please refer to W1-3-2. And analysis in Sec. 5.1 also suggests that tailored training data can help the student model learn to reason.
>    - **As the negative samples for self-reflection learning**, which contributes to enhancing the reasoning performance of student LM (c.f. Sec. 5.2). We also add more details about it in W1 of Reviewer MSxv.
>
> 2. We need to clarify that the **hint (golden answer) is often added to improve the teacher LLM's outputs, not associated with student feedback** (see Fig. 3).  Thus, we think results in Table 2 can verify the contribution of student feedback. And case study in Table 3 and Appendix C also indicates that the generated rationales w/ student feedback are better for student's current learning status.
>
> 3.  **Setting the same request quota instead of the number of training data is fairer to compare the efficacy of the methods.** In Table 2, we set the same request quota to teacher LLM, and find adding student feedback can collect more valid training data than w/o, which commonly leads to better training. **Preparing the student feedback only needs the smaller LM to infer over the training set, which is relatively costless compared to training.**
>
> 4.  **To extend the findings from Table 2 to multiple rounds**, we update the preliminary results of 3rd round, based on different 2nd checkpoints which also means a different number of requests to teacher LLM. **Our method can often fine-tune student LM to achieve better performance while using less request quota, saving API costs.**
>
>    | Dataset      | # Request | # Success | Accuracy |
>    | ------------ | --------- | --------- | -------- |
>    | GSM8k        | 4947      | 4434      | 30.6     |
>    | w/o feedback | 5113      | 3890      | 28.2     |
>    | SVAMP        | 117       | 96        | 52.3     |
>    | w/o feedback | 125       | 99        | 47.2     |
>    | StrategyQA   | 82        | 79        | 64.6     |
>    | w/o feedback | 116       | 87        | 63.9     |
>
> ### W3: only decoder-only models are tested
>
> There are three main reasons for adopting decoder-only (GPT-like) models to validate our method:
>
> 1. **Decode-only models recently gained substantial attention from both research and open-source communities**, such as ChatGPT, LLaMA, MPT, and Falcon. With limited resources, there exists a preference for us to choose widely adopted model architectures.
>
> 2. **There are no appropriate model sizes of T5 series for our resources**, because:
>
>    - Recent research and practice from the open-source community suggest models with about 7B parameters have the potential to be capable of emergent abilities, such as instruction-following and reasoning.
>    -  Our GPU resources cannot afford T5-xxl (11B, nearly 2x larger than the adopted GPT-j-6B).
>
> 3. **The proposed method does not rely on any specific features of decoder-only or encoder-decoder architectures**, thus it should be effective for different LMs. We also take a feasibility study in Sec. 5.4 to validate its effectiveness with the other 3 models. We would like to add results of T5-large-760M to feasibility study in Sec. 5.4.  The preliminary results on SVAMP and StrategyQA datasets are in the following table. We find that our self-reflection learning can effectively help the student model T5 in reasoning performance. The effect of multi-round learning is limited due to student's relatively small capacity.
>
>    | Method / Dataset     | SVAMP | StrategyQA |
>    | -------------------- | ----- | ---------- |
>    | Student T5 (initial) | 0.0   | 0.0        |
>    | +Distillation        | 11.0  | 62.0       |
>    | +Self-Reflection     | 14.7  | 64.0       |
>    | +Multi-round         | 15.3  | 64.8       |
>
> ### W4: human evaluation
>
> 1. Human evaluation is valuable to verify the enhanced reasoning ability of student LMs. However, due to the limited budget and resources, **it is unaffordable for us to take qualified human evaluations.**
> 2. Recently there has been a line of research[1,2,3] showing GPT-4 could be an alternative evaluator to human evaluation. **Thus, we decide to use GPT-4**, the currently most capable and accessible large language model **to automatically evaluate the reasoning ability of our student LMs**.
> 3. We randomly sample 50 samples (100 in total) for correct and wrong rationales of each dataset, respectively. Following the format of Table 4, we show the GPT-4 scores of the student LM in different rounds of learning.  **It shows that there is an enhancement in the quality of generated rationales as the learning rounds progress.** We would like to add these results to strengthen our paper.
> 4. **Details for GPT-4 evaluation.** We use a 5-shot prompt to instruct GPT-4 not only to rate each sample from 1 to 5, but also to provide justification of the score. The evaluation metrics are based on the accuracy and quality of generated rationales and answers. The score of each sample is averaged over 4 samplings from GPT-4.
>
> | Dataset | Round   | Score of Correct | Score of Wrong |
> | ------- | ------- | ---------------- | -------------- |
> | GSM8K   | Initial | 2.59 ± 0.27      | 1.02 ± 0.07    |
> |         | 1       | 4.50 ± 0.18      | 1.15 ± 0.20    |
> |         | 2       | 4.88 ± 0.14      | 1.26 ± 0.23    |
> | SVAMP   | Initial | 4.53 ± 0.20      | 1.07 ± 0.18    |
> |         | 1       | 4.86 ± 0.16      | 1.09 ± 0.21    |
> |         | 2       | 4.90 ± 0.24      | 1.11 ± 0.20    |
> | CSQA    | Initial | 4.44 ± 0.22      | 1.24 ± 0.28    |
> |         | 1       | 4.84 ± 0.27      | 1.41 ± 0.28    |
> |         | 2       | 4.96 ± 0.12      | 1.55 ± 0.33    |
>
> [1] Fu, Jinlan, et al. "GPT Score: Evaluate as you desire." *arXiv preprint arXiv:2302.04166* (2023).
>
> [2] Cheng-Han Chiang and Hung-yi Lee. 2023. [Can Large Language Models Be an Alternative to Human Evaluations?](https://aclanthology.org/2023.acl-long.870). In *Proceedings of the 61st Annual Meeting of the Association for Computational Linguistics (Volume 1: Long Papers)*, pages 15607–15631, Toronto, Canada. Association for Computational Linguistics.
>
> [3] Hackl, Veronika, et al. "Is GPT-4 a reliable rater? Evaluating Consistency in GPT-4 Text Ratings." *arXiv preprint arXiv:2308.02575* (2023).

---

### Official Review · Reviewer_MSxv · 2023-08-04

**Soundness:** 4

**Excitement:**

4: Strong: This paper deepens the understanding of some phenomenon or lowers the barriers to an existing research direction.

**Paper Topic And Main Contributions:**

This work proposes to train smaller LMs with rationales generated by large language models (LLMs) to enable reasoning ability of smaller LMs.

Compared to previous works, this work introduces the multi-round interactive learning where the student also provides its learning status to the teacher LLM to get better rationales leading to improved training. Furthermore, this method includes self-reflection which guide small LMs to correct the errors itself.

In experiments, based on GPT-J with 6B parameters, the proposed method improves the reasoning ability of small language models in arithmetic and commonsense reasoning tasks.

**Reasons To Accept:**

The paper is well-structured and effectively conveys its ideas and experimental results.
The proposed idea is novel enough, particularly the concept of utilizing student feedback to improve the rationales provided by the LLM, which has not been explored in previous works.

The experimental results and ablation studies convincingly demonstrate the method’s efficacy in enhancing performance.

**Reasons To Reject:**

There is no major weakness.

One minor concern is about the effectiveness of self-reflection. The clarification is needed on how Equation 2 enables small models to generate accurate rationales, as it merely seeks to minimize the distance between representations of the ground-truth and correct rationales than wrong rationales? Is it helpful to language model also in inference stage so that the language model generates correct rationale? Incorporating latnet space visualization could be helpful in comprehending the role of self-reflection.

**Reproducibility:**

4: Could mostly reproduce the results, but there may be some variation because of sample variance or minor variations in their interpretation of the protocol or method.

**Reviewer Confidence:**

4: Quite sure. I tried to check the important points carefully. It's unlikely, though conceivable, that I missed something that should affect my ratings.

---

> ### Author Rebuttal · Authors · 2023-08-29
>
> Thanks for your appreciation of our work. We are happy to discuss the effectiveness of self-reflection and incorporate the suggested visualizations to comprehend the role of self-reflection learning.
>
> ### W1: clarification of effectiveness of self-reflection learning
>
> 1. Yes, self-reflection learning (Eq. 2) seeks to minimize the distance between representations of the ground truth and correct rationales than wrong rationales.
>    - The self-reflection learning is essentially a form of contrastive learning, which is shown to be effective in **alleviating the problem of anisotropic distribution of token representations** in neural text generation[1].
>    - Also, the effect is kind of **aligning the preference of the student model with the teacher LLM's reasoning**, which is similar to alignment with human values[2,3].
> 2. An intuitive explanation is that extending the distance between correct and wrong rationales **may help the student model learn better representations of correct rationales** (i.e., help learn from the teacher LLM). And from experimental results in the Table 1 (+ Self-Reflection) and Figure 4 ($\lambda > 0.0$), we empirically find that it could help student model generate better rationales in inference.
> 3. Thanks for your advice on visualizing the latent space to help validate its effectiveness.
>    - We first randomly sample $100$ questions with their corresponding correct (from teacher's feedback) and wrong (from student's mistakes) rationales.
>    - We can encode each question-rationale pairs ($LM([x, r_\text{correct}])$ and $LM([x, r_\text{wrong}])$) using their latent space representations.
>    - We adopt t-SNE to visualize these high dimensional vector representations. From the visualizations, we find that self-reflection learning can effectively **cluster correct rationales and wrong rationales respectively**, helping the model to distinguish each other.
> 4. **Due to the limits of inserting figures in OpenReview platform**, we instead show the L2 distance and $\log \frac{L([x, r_\text{correct}])}{L([x, r_\text{wrong}])}$in the following table, where $L$ denotes the likelihood of student model, and represents for the preference between correct and wrong rationales. The results show that self-reflection learning can help the student model to better distinguish correct rationales from wrong ones.  **We will incorporate the visualizations in the camera ready version.**
>
> | Dataset    | Metric                                | Student Model w/o Self-reflection | Student Model w/ Self-reflection |
> | ---------- | ------------------------------------- | --------------------------------- | :------------------------------- |
> | GSM8K      | L2(correct, wrong) $\uparrow$         | 51.00                             | 65.08                            |
> | GSM8K      | Log(L(correct) / L(wrong)) $\uparrow$ | 73.63                             | 79.11                            |
> | StrategyQA | L2(correct, wrong) $\uparrow$         | 5.03                              | 24.78                            |
> | StrategyQA | Log(L(correct) / L(wrong)) $\uparrow$ | 96.54                             | 98.91                            |
>
> [1] Su, Yixuan, et al. "A contrastive framework for neural text generation." *Advances in Neural Information Processing Systems* 35 (2022): 21548-21561.
>
> [2] Rafailov, Rafael, et al. "Direct preference optimization: Your language model is secretly a reward model." *arXiv preprint arXiv:2305.18290* (2023).
>
> [3] Song, Feifan, et al. "Preference ranking optimization for human alignment." *arXiv preprint arXiv:2306.17492* (2023).

---

### Meta-Review · Area_Chair_Nnhs · 2023-09-16

**Recommendation:** 4

**Metareview:**

This paper proposes generating rationales from bigger LLMs to help smaller LLMs come to the right decision. The discussion period raised several important weaknesses of the approach: it relies on the correctness of the final answer and doesn't have a human evaluation. Despite these weaknesses, the reviewers seem to agree that this approach is not only practically useful for creating interpretable distilled models but can also improve our understanding of what larger LLMs have to offer over smaller ones.

---

### Decision · Program_Chairs · 2023-10-07

**Decision:**

Accept-Main

**Comment:**

This paper proposes generating rationales from bigger LLMs to help smaller LLMs come to the right decision. The discussion period raised several important weaknesses of the approach: it relies on the correctness of the final answer and doesn't have a human evaluation. Despite these weaknesses, the reviewers seem to agree that this approach is not only practically useful for creating interpretable distilled models but can also improve our understanding of what larger LLMs have to offer over smaller ones.